# The Effect of Audio and Audiovisual Distraction on Pain and Anxiety in Patients Receiving Outpatient Perineal Prostate Biopsies: A Prospective Randomized Controlled Study

**DOI:** 10.3390/cancers17060959

**Published:** 2025-03-12

**Authors:** Julia Carola Kaulfuss, Nicolas Hertzsprung, Henning Plage, Benedikt Gerdes, Sarah Weinberger, Thorsten Schlomm, Maximilian Reimann

**Affiliations:** Department of Urology, Charité-Universitätsmedizin Berlin, Corporate Member of Freie Universität Berlin and Humboldt-Universität zu Berlin, Hindenburgdamm 30, 12203 Berlin, Germany; nicolas.hertzsprung@charite.de (N.H.); maximilian.reimann@charite.de (M.R.)

**Keywords:** prostate cancer, audiovisual distraction, perineal prostate biopsy, novel therapies, urological cancer, cancer diagnostics

## Abstract

Prostate biopsies are a common medical procedure to diagnose prostate cancer, but can cause significant anxiety and discomfort for patients. To address this, we investigated whether audio or audiovisual distraction can help to reduce pain, anxiety and stress during the procedure. Our study included 168 male patients undergoing prostate biopsies, divided into three groups: one group received no intervention, the audio distraction group listened to music via headphones, and the audiovisual distraction group received audiovisual glasses to watch documentaries of their interest. We found that audiovisual distraction significantly reduced the subjective perception of the duration of the procedure. However, audiovisual distraction did not significantly lower pain or anxiety levels compared to the control group. Even so, these findings highlight the potential of audiovisual distraction as a simple, cost-effective way to improve patient comfort during outpatient procedures, with implications for broader investigation in outpatient procedures.

## 1. Introduction

Outpatient procedures (OPs) are an essential part of diagnostic and therapeutic surgical departments, as they reduce lengths of hospitalization and have the potential to be more cost-efficient than inpatient procedures [1]. As the gold standard in prostate cancer diagnostics, prostate biopsies, perineal or transrectal, are an OP which is performed more than 70,000 times/year in Germany [2].

Perineal prostate biopsies (PPBs) are invasive OPs which can cause distress, anxiety and enhanced pain in patients. Studies showed that 20% of men feel highly stressed about the procedure, while some of them cannot tolerate it without local anesthesia [1,3,4]. This may lead to increased demand for pain killers, local anesthesia, and sometimes early termination of the procedure or procedure postponement due to the need for general anesthesia. Reducing anxiety and pain in patients receiving PPBs by using additive, non-pharmacological tools like audio and audiovisual distractions can easily improve patient care while keeping costs and administration efforts low [5].

Audio distraction has already been found to be an effective, low-cost and non-pharmacological tool to decrease anxiety and pain in various medical procedures and interventions [6,7,8]. Several studies have underlined the positive effects of audio distraction on pain perception and anxiety in urological procedures such as prostate biopsies, cystoscopy, shock wave lithotripsy or urodynamic studies [7,9,10,11]. Concerning that, studies that examined the effects of audio distraction on pain and anxiety in patients receiving prostate biopsies have only included transrectal biopsies, but not PPBs [12,13,14,15]. On the contrary, the number of PPBs is rising due to lower complication rates (e.g., infections, bleeding) and consistent accuracy compared to transrectal prostate biopsies [16]. Consequently, our study aims to evaluate the effects of audio and audiovisual distractions on PBBs.

Moreover, medical researchers have already examined audiovisual distraction as a useful tool to reduce pain perception, anxiety and demands for sedation medication [10,17,18,19,20]. The neurobiological background is the attention shift towards pleasing audiovisual stimuli, which reduce the activation capacity of brain areas processing anxiety, pain and stress [21]. Audiovisual distractions showed positive effects on pain perception in urological procedures like cystoscopy and shock wave lithotripsy [10,17]. To our knowledge, there are no studies that have yet addressed the effects of audiovisual distraction on PPB.

In summary, the number of OPs in urology is rising, and, likewise, the need for alternative tools to reduce pain, stress and anxiety in patients receiving procedures under local anesthesia is also increasing. Especially for patients who feel stressed about outpatient procedures performed with local anesthesia, distraction tools can help to improve patient comfort and thereby lower levels of local anesthesia and decrease the need of general anesthesia or inpatient care. Audio and audiovisual distraction tools are low-cost and promising strategies to lower pain perception and anxiety in patients, and have not been previously examined in relation to PPBs, one of the most common OP in urology.

The aim of this study was to examine the effects of audio and audiovisual distraction with objective and subjective parameters representing pain and anxiety levels of patients undergoing outpatient PPBs.

## 2. Materials and Methods

Ethical approval was received from the ethical board of Charité-Universitätsmedizin Berlin, including the standards of the institutional research committee for all participants and procedures performed in this study. The study was registered in the German Clinical Trials Registry (DRKS) on 26 January 2022 (DRKS00025602). The registry fulfills the requirements of the WHO International Clinical Trial Registry Platform. Informed consent was obtained from all participants.

For the present study, we recruited male patients over 18 years old who underwent a PPB using EXACT Imaging micro-ultrasound (Exact Imaging, Markham, Canada) for prostate cancer diagnostics between November 2021 and November 2022 at Charité-Universitätsmedizin Berlin, Germany. The participants were block randomized with a maximum size of 5 participants per day into three groups: a control group (CG), an audio distraction group (ADG) and an audiovisual distraction group (AVDG). The CG got no intervention, while the ADG received headphones (V7 Over-Ear Headphones HA701-3EP; Dornach/München, Germany), with a standard selection of classical music, and the AVDG was provided with audiovisual glasses (Happymed audiovisual glasses; Happymed, Vienna, Austria) while the PPB was performed. Patients were asked to choose from a 10-piece documentary film selection, which was then played on the audiovisual glasses.

The primary endpoint of this study was the perception of pain during PPB, as assessed by the Numeric Rating Scale (NRS) and the Visual Analogue Scale (VAS). Patients completed NRS and VAS questionnaires regarding three time points: before the PPB, during the procedure, after 1 min after the PPB. Secondary endpoints to evaluate pain perception were objective parameters like heart and blood pressure. In detail, heart rate and systolic blood pressure were measured before, during, and 1 min after the PPB. We did not pause any pharmacological medication (e.g., beta blockers) that could influence those parameters. For data analysis, the difference between the maximum heart rate during the procedure and the heart rate at baseline was calculated, and the same was done for blood pressure.

To examine anxiety and stress, we used the validated questionnaire State-Trait Anxiety Inventory (STAI) [22,23] and salivary cortisol [24]. Cortisol levels have been found to be higher in stressful periods or situations [25]. To test cortisol levels in salvia, patients received a cotton test pad to chew on 5 min before and 30 min after the procedure.

Additionally, we measured subjective and objective procedure time (SPT/OPT), monitored in minutes. SPT was defined as the subjective duration of the procedure, perceived individually by each patient, while OPT was actual time measured with a stopwatch. The difference between SPT and OPT was calculated by subtracting the subjective time from the objective time.

We did not assess pharmacological drugs or baseline levels of anxiety and pain that could influence vital parameters, STAI scores or cortisol levels.

Regarding active surveillance strategies, with the need for yearly performed prostate biopsies in mind, we assessed patients’ willingness to repeat the procedure [26].

All surgeons are fully trained urologists equally experienced in performing PPBs regularly. PPBs were performed using a unique local anesthesia sequence of the department of urology at Charité-Universitätsmedizin Berlin of 20 mL (1:1, lidocaine 1%: NaCl) in cutaneous area followed by a deep tissue anesthesia to the left, right and in front of the prostate, with 10 mL pure lidocaine 1%. To specify, the superficial local anesthesia administration was performed with a single injection, followed by a superficial infiltration of the tissue by moving the needle subcutaneously. In this way, the patient, ideally, perceived only a single puncture as painful. The PPB was performed with 10 biopsies in a standard international scheme of the prostate with an additional three biopsies of the target lesion.

We performed a power analysis based on the results of a pilot study which defined a necessary sample size of at least 50 patients per group. The following statistical analyses were performed using SPSS, version 29 (IBM, Armonk, NY, USA). Graphs were also visualized using SPSS, version 29, and edited with PowerPoint, version 16 (Microsoft, Redmond, WA, USA). Data and tables were managed with Excel, version 16 (Microsoft, Redmond, WA, USA). Descriptive statistics were given as mean with standard deviation. For the statistical evaluation, the results of the three groups were compared to each other using a *t*-test. To study the three groups in terms of the distribution of the willingness to repeat the procedure, the chi-square test was applied. All tests were two-sided, and *p* < 0.05 was considered to indicate statistical significance.

## 3. Results

A total of 168 male patients were included in this study. Of these, 62 were randomized to the CG, 62 to the ADG, and 44 to the AVDG (Table 1).

For the primary endpoint pain perception, represented in NRS and VAS, we did not find a significant difference between the three groups (Figure 1).

Objective pain perception parameters like systolic blood pressure (CG: *n* = 62, 7.74 ± 13.98 mmHg; ADG: *n* = 60, 4.03 ± 16.23 mmHg; AVDG: *n* = 43, 4.84 ± 11.86 mmHg) and heart rate (CG: *n* = 62, 2.18 ± 15.28 bpm; ADG: *n* = 60, −1.07 ± 18.49 bpm; AVDG: *n* = 41, 6.54 ± 12.26 bpm) did not differ significantly between the three groups.

The assessment of stress and anxiety in the procedure using cortisol in saliva and STAI scores also showed no significant difference between the three groups (Figure 2).

The willingness to repeat the procedure was low. Overall, 60.1% of the participants stated that they would be prepared to repeat the procedure as it was carried out. In the CG (*n* = 56) the willingness-to-repeat rate was 76.79%, in the ADG (*n* = 58) 58.62% and in the AVDG (*n* = 39) 61.54%.

Furthermore, there was no difference in OPT (CG: *n* = 59, 15.00 ± 4.67 min; ADG: *n* = 60, 14.85 ± 5.73 min; AVDG: *n* = 39, 13.41 ± 4.22 min), but we examined significantly shorter SPT and a lower SPT/OPT ratio in the AVDG compared to the CG (Figure 3).

## 4. Discussion

The aim of the present study was to investigate the effects of audio and audiovisual distraction on pain perception and anxiety in patients receiving PPBs as a common and representative outpatient procedure in urology.

First, we discovered a significant effect of audiovisual distraction on the SPT of patients receiving PPBs. More precisely, patients who had audiovisual distractions (AVDG) documented significantly shorter SPTs than patients who received audio distraction alone (ADG) or no distraction (CG) (Figure 1). This effect can also be observed when considering SPT in relation to OPT (Figure 2). Former studies examined that audio distraction alone did not shorten the OPT of patients receiving (rectal) prostate biopsies. However, SPT was not observed in these studies [12,15]. As audiovisual distraction shifts the attention from brain areas processing anxiety to pleasing audiovisual stimuli; it is likely that this distraction may weaken the patient’s sense of time, and thereby reduce SPT [21]. As we specifically asked our patients to choose a documentary they want to watch and are interested in, the grade of distraction may even be enhanced. Concerning that, studies performed in psychological research have already examined the relationship between subjective time overestimation and perceiving external unpleasant stimuli. To put the findings precisely, patients tend to experience time as being longer when in pain [27,28,29]. Taking that into account, we summarize that even though SPT is a highly subjective parameter in our study, it may indirectly represent reduced pain and anxiety in patients receiving audiovisual distractions during PPBs.

Second, we found no significant effect of audio or audiovisual distractions on heart rate, blood pressure or salivary cortisol. This data differs from other studies, which have found a significant positive effect of audio distraction on heart rate and blood pressure during transrectal prostate biopsies [14,15]. To our knowledge, other studies did not adjust pharmacological medication either. Thus, the influence of pharmacological medication should be considered in following studies. Studies before did not test salivary cortisol levels.

Third, we did not examine a significant difference between the groups concerning NRS and VAS representing pain perception. In former research, four studies reported mixed findings on the effects of audio distraction on pain perception as represented in the VAS scores of patients receiving rectal prostate biopsies, whereas a large randomized trial of Packiam et al. showed no significant effects of audio distraction [12,13,14,15,30]. To our knowledge, there are no studies that have examined the effect of audio distraction on PPB. Former studies did examine positive effects of audiovisual distraction on pain perception in patients receiving urological procedures like cystoscopy and shock wave lithotripsy [10,17], but no studies have yet addressed perineal or rectal prostate biopsies.

Forth, scores in STAI questionnaires did not show any significant difference between the groups in our study. Some studies did examine significant differences in STAI scores of patients receiving audio distraction during transrectal prostate biopsy whereas others did not [14,15,30].

Fifth, we did not find any difference between the groups in terms of patients’ willingness to repeat the procedure.

Our study is not devoid of limitations. We did not assess pharmacological drugs for high blood pressure or tachycardia which could influence the measured vital parameters. We also did not assess the baseline levels of anxiety and pain, for example, chronical pain, or anxiety disorders, which could influence STAI scores or cortisol levels. Further studies should take this into account. Moreover, our study only includes male participants and focusses on one OP procedure. For further research, a more heterogenous participant cohort could be interesting, along with to examining possible sex-specific differences in the effects of audiovisual distraction on pain and anxiety. Nevertheless, yet there is no other study which has yet examined the effects of audio and audiovisual distractions on patients receiving PPBs [31]. Also, other distraction tools, such as virtual reality hypnosis, should be investigated further [32,33].

## 5. Conclusions

Our study highlights the potential of audiovisual distraction as a non-pharmacological tool to improve patient comfort during outpatient PPBs. While audiovisual distraction did not significantly reduce pain or anxiety levels, it notably shortened the subjective perception of procedure time. This finding suggests that audiovisual distraction may shift patients’ focus away from the invasive nature of the procedure, creating a more tolerable experience. These results underline the importance of exploring simple, cost-effective strategies to enhance patient care and thereby compliance during invasive procedures. Further research is needed to investigate the effects of audio and, especially, audiovisual distraction over a broader range of outpatient procedures, as well as its impact on more diverse patient populations. This could enable integration of audiovisual distraction tools into routine clinical practice to improve the quality of patient care in surgical departments.

## Figures and Tables

**Figure 1 cancers-17-00959-f001:**
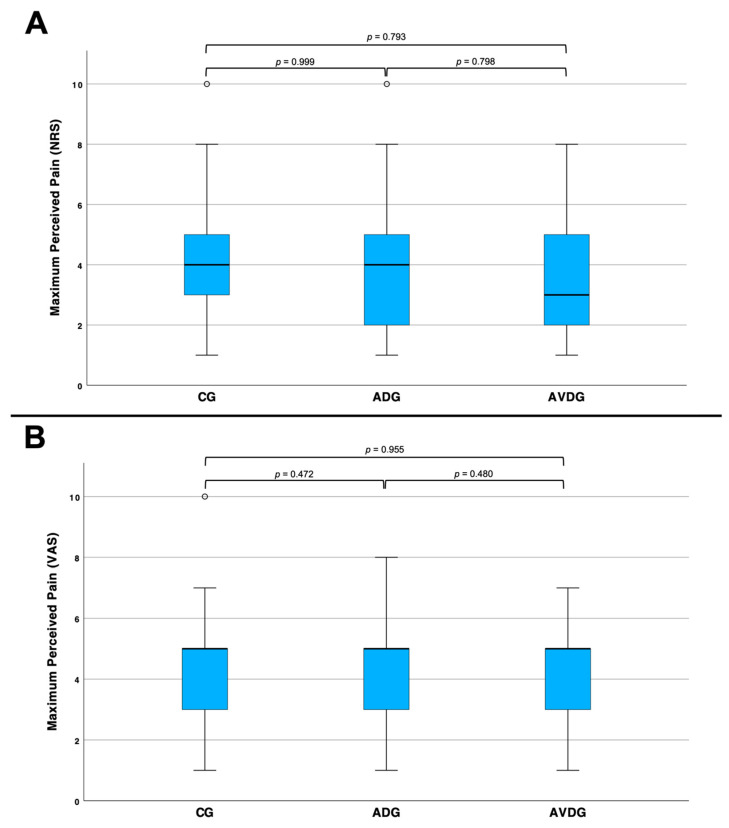
Comparison across all three groups in terms of the most severe pain during the biopsy as a boxplot for NRS (**A**) and VAS (**B**). (**A**): the patients in the CG (*n* = 61) reported pain levels of 4.03 ± 2.00 points, in mean value, on the NRS, those in the ADG (*n* = 60) indicated pain levels of 4.03 ± 2.10 points and those in the AVDG (*n* = 40) reported mean pain levels of 3.93 ± 2.03 points. There is no significant difference between the three groups. (**B**): the patients in the CG (*n* = 47) reported mean pain levels of 4.09 ± 1.86 points on the VAS, those in the ADG (*n* = 49) indicated pain levels of 4.35 ± 1.69 points and those in AVDG (*n* = 33) reported mean pain levels of 4.06 ± 1.94 points. There is no significant difference between the three groups.

**Figure 2 cancers-17-00959-f002:**
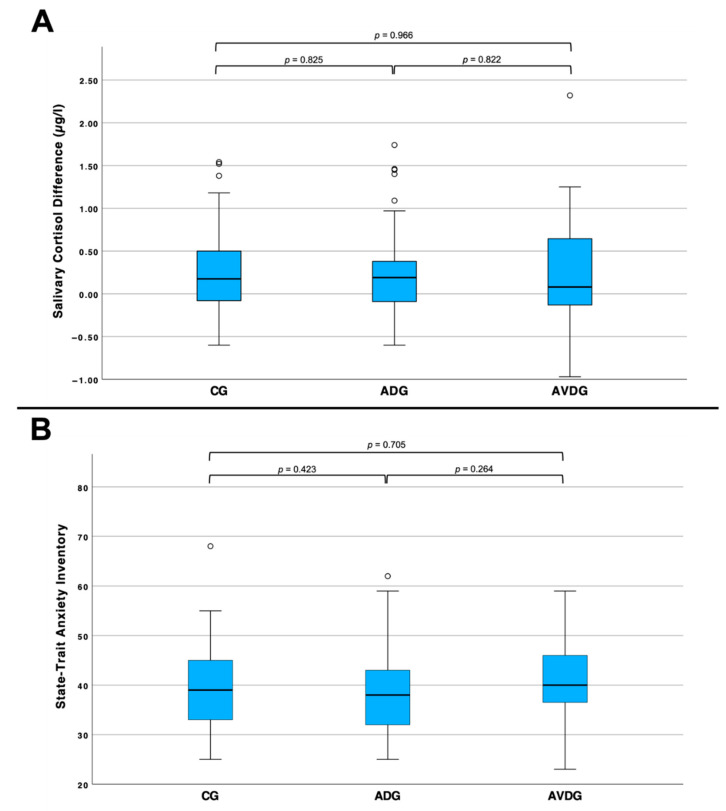
Comparison across all three groups in terms of the examined stress (**A**) and anxiety (**B**) levels. (**A**): patients of the CG (*n* = 42) had an increase of 0.26 ± 0.54 µg/L in salvia cortisol, those in the ADG (*n* = 42) increased by 0.24 ± 0.56 µg/L and those in the AVDG (*n* = 40) saw an increase of 0.27 ± 0.67 µg/L in mean salvia cortisol. There is no significant difference between the three groups. (**B**): the patients in the CG (*n* = 61) reported mean anxiety levels of of 39.80 ± 8.22 points on STAI, those in the ADG (*n* = 62) reached 38.60 ± 8.41 points and those in the AVDG (*n* = 44) reported 40.41 ± 7.84 points in STAI. There is no significant difference between the three groups.

**Figure 3 cancers-17-00959-f003:**
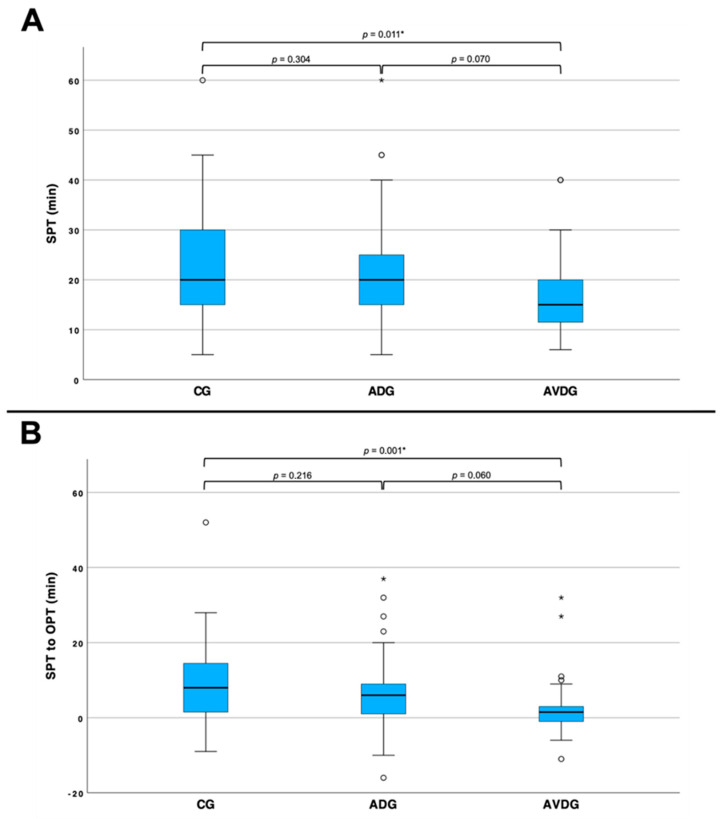
Comparison across all three groups in terms of SPT (**A**) and SPT/OPT (**B**). (**A**): the patients in the CG (*n* = 58) reported a subjective duration of 23.31 ± 9.06 min for the procedure, those in the ADG (*n* = 58) reported 21.05 ± 10.45 min and those in the AVDG (*n* = 39) reported 17.26 ± 8.22 min in mean subjective time. There is a significant difference between the CG and the AVDG. (**B**): the patients in the CG (*n* = 55) had a mean difference of 8.27 ± 10.28 min between SPT and OPT, those in the ADG (*n* = 57) had 6.37 ± 9.22 min and those in the AVDG (*n* = 34) had a mean difference of 2.85 ± 8.18 min. There is a significant difference between the CG and AVDG. “*” next to the p-values just indicates it to be significant “*” in the diagram represents data marks outside of standard deviation (outliers).

**Table 1 cancers-17-00959-t001:** Table showing the number and age of patients overall and per group.

	Total	CG	ADG	AVDG
Number	168	62	62	44
Age	67.88 ± 8.05	68.71 ± 1.16	66.89 ± 0.92	68.11 ± 1.15

## Data Availability

The raw data supporting the conclusions of this article will be made available by the authors on request.

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
