# Peer review of "The Effect of Audio and Audiovisual Distraction on Pain and Anxiety in Patients Receiving Outpatient Perineal Prostate Biopsies: A Prospective Randomized Controlled Study"

_cancers, 2025, doi:10.3390/cancers17060959_

Round 1
Reviewer 1 Report
Comments and Suggestions for Authors
page 1, line 36 - there is an abbreviation that has no corresponding text: CT. I think CT should be replaced with CG.
page 3 - lines 122-126 - I suggest you to further detail the method of performing the local anesthesia. In my experience, the very moment of injecting the anesthetic can be a source of pain.
I congratulate you for the idea - it is simple and can bring benefits. It would be an argument in favor of the method whose opponents argue that transperineal biopsy, recommended by the guidelines, causes pain and cannot be practiced with local anesthesia.
Author Response
Comment 1: page 1, line 36 - there is an abbreviation that has no corresponding text: CT. I think CT should be replaced with CG.
Response 1: Thank you very much for pointing this out. We totally agree. "CT" is now replaced with "CG" (line 37)
Comment 2: page 3 - lines 122-126 - I suggest you to further detail the method of performing the local anesthesia. In my experience, the very moment of injecting the anesthetic can be a source of pain.
Response 2: Agree. We now tried to explain further the method of performing local anesthesia by adding the following: "To specify, the superficial local anesthesia was performed through a single injection, followed by a superficial infiltration of the tissue by moving the needle subcutaneously. In this way, the patient ideally perceived only a single puncture as painful." (lines 131-134, Page 3)
Comment 3: I congratulate you for the idea - it is simple and can bring benefits. It would be an argument in favor of the method whose opponents argue that transperineal biopsy, recommended by the guidelines, causes pain and cannot be practiced with local anesthesia.
Response 3: Thank you so much for your comment and revision.
Reviewer 2 Report
Comments and Suggestions for Authors
The Authors carried out a randomized study in order to identify effects of audio- and audiovisual distraction on pain and anxiety of patients receiving perineal prostate biopsy (PPB).
The abstract is easy to understand and short. It explains how we can use audiovisual distraction to improve patient comfort during outpatient perineal prostate biopsy (PPB). Adding a sentence about the wider use of distraction techniques in urological procedures would make the abstract stronger.
The introduction explain in a good way the background on PPB and the need for non-pharmacological interventions to improve patient experience. It gives all the information needed. Some references could be more explicitly integrated to support the discussion on the limitations of existing pain management strategies.
Methods section is detailed, well describing the study design. It would be beneficial to clarify whether potential confounders (e.g., prior pain experiences, anxiety disorders, pharmacological treatments) were taken into account during the analysis.
The results are presented in a clear and concise manner, with significant findings highlighted for ease of reference. The statistical analysis is thorough, however, the tables could be enhanced by including more detailed legends to explain abbreviations and statistical tests. The primary endpoint analysis (pain perception) did not demonstrate a significant difference between the groups, which aligns with previous studies on distraction techniques. However, the substantial reduction in SPT in the audiovisual distraction group warrants further discussion on its potential clinical relevance.
The discussion is clearly formulated. However, the author should provide an analysis of why pain and anxiety levels did not significantly decrease despite changes in subjective time perception. Furthermore, the author should explain the controversy surrounding non-pharmacological pain management techniques and their variable effectiveness.
In the field of future research, recommendations are appropriate, but could be expanded to explore alternative distraction techniques, such as virtual reality hypnosis. There are already published articles on this topic.
Author Response
Comment 1: The Authors carried out a randomized study in order to identify effects of audio- and audiovisual distraction on pain and anxiety of patients receiving perineal prostate biopsy (PPB). The abstract is easy to understand and short. It explains how we can use audiovisual distraction to improve patient comfort during outpatient perineal prostate biopsy (PPB). Adding a sentence about the wider use of distraction techniques in urological procedures would make the abstract stronger.
Response 1: Thank you for your comment. To improve our abstract, we included to following sentences: "Audiovisual distraction tools already showed positive effects on pain perception in some urological procedures" (lines 23-24) and "To accelerate the wider implementation of audiovisual distraction as a cost-efficient tool in outpatient urological procedures, further studies should examine its effect on different OPs with a more heterogeneous patient group." (lines 40-42)
Comment 2: The introduction explain in a good way the background on PPB and the need for non-pharmacological interventions to improve patient experience. It gives all the information needed. Some references could be more explicitly integrated to support the discussion on the limitations of existing pain management strategies.
Response 2: Agree. We added the following to state more explicitly why the existing pain management strategies are limited: "Especially for patients who feel stressed about outpatient procedures performed with local anesthesia, distraction tools can help to improve patient comfort, to lower general anesthesia or inpatient care." (lines 79-81)
Comment 3: Methods section is detailed, well describing the study design. It would be beneficial to clarify whether potential confounders (e.g., prior pain experiences, anxiety disorders, pharmacological treatments) were taken into account during the analysis.
Response 3: We agree with that comment. We adressed that only in our limitation section: "We did not assess pharmacological drugs for high blood pressure or tachycardia which could influence the measured vital parameters. We also did not assess the baseline levels of anxiety and pain for example chronical pain, or anxiety disorders which could influence STAI scores or cortisol levels." (lines 235-238). Additionally, we now included it in the methods section by adding the following: "We did not assess pharmacological drugs or baseline levels of anxiety and pain that could influence vital parameters, STAI scores or cortisol levels." (lines 123-124)
Comment 4: The results are presented in a clear and concise manner, with significant findings highlighted for ease of reference. The statistical analysis is thorough, however, the tables could be enhanced by including more detailed legends to explain abbreviations and statistical tests. The primary endpoint analysis (pain perception) did not demonstrate a significant difference between the groups, which aligns with previous studies on distraction techniques. However, the substantial reduction in SPT in the audiovisual distraction group warrants further discussion on its potential clinical relevance.
Response 4: Thank you very much for your comment. We tried to keep the figures simple and clear and provide the details in the discription below. We hope that those details explain results and abbreviations concisely.
Comment 5: The discussion is clearly formulated. However, the author should provide an analysis of why pain and anxiety levels did not significantly decrease despite changes in subjective time perception. Furthermore, the author should explain the controversy surrounding non-pharmacological pain management techniques and their variable effectiveness. In the field of future research, recommendations are appropriate, but could be expanded to explore alternative distraction techniques, such as virtual reality hypnosis. There are already published articles on this topic.
Response 5: We cannot exactly explain why subjective procedure times changed while pain and anxiety scales did not significantly decrease. Therefor we tried to provide an explanation how audiovisual distraction can shorten subjective time perception. In the field of future research, we totally agree that alternative distraction techniques shoult be explored. Therefor, we added the following to our manuscript: "Also, other distraction tools, such as virtual reality hypnosis should be investigated further (32,33)." (lines 242-244) Ref: 32. Perenic E, Grember E, Bassard S, Koutlidis N. Impact of virtual reality on pain management in transrectal MRI-guided prostate biopsy. Front Pain Res (Lausanne). 2023;4:1156463 33. Wong J, McGuffin M, Smith M, Loblaw DA. The use of virtual reality hypnosis for prostate cancer patients during transperineal biopsy/gold seed implantation: A needs assessment study. J Med Imaging Radiat Sci. 2023;54(3):429-35. (lines 332-335)
Reviewer 3 Report
Comments and Suggestions for Authors
I read with interest this study about the impacts of audio and audiovisual distraction on pain and anxiety of patients receiving perineal prostate biopsies (PB). The paper is well written and the author explain the problems related with the diagnosis of this "difficult" disease. Figures, the writing style, design, interpretations, and quality of the work are satisfying. References are lacking.
A diagnosis of prostate cancer is associated with increased patient anxiety. Many authors prove that the negative impact of PB on patient well-being can even begin while waiting for the programmed procedure and that there is increased anxiety and worry related immediately before undergoing biopsy with a minority (20%) suffering high levels of distress at this time. Several studies used the Hospital Anxiety and Depression Scale, which are created to detect clinical cases of depression and anxiety.
Can the authors report and expand these considerations in a discussion section? (I suggest adding the reference: Impact of music on anxiety and pain perception among men undergoing prostate biopsy: Synthesis of qualitative literature. Dell’Atti L. et al. Complement Ther Clin Pract 2021; 43:101330. doi: 10.1016/j.ctcp.2021.101330).
Author Response
Comment 1: I read with interest this study about the impacts of audio and audiovisual distraction on pain and anxiety of patients receiving perineal prostate biopsies (PB). The paper is well written and the author explain the problems related with the diagnosis of this "difficult" disease. Figures, the writing style, design, interpretations, and quality of the work are satisfying. References are lacking.
Response 1: Thank you very much for your revision. We will try to improve the lacking references.
Comment 2: A diagnosis of prostate cancer is associated with increased patient anxiety. Many authors prove that the negative impact of PB on patient well-being can even begin while waiting for the programmed procedure and that there is increased anxiety and worry related immediately before undergoing biopsy with a minority (20%) suffering high levels of distress at this time. Several studies used the Hospital Anxiety and Depression Scale, which are created to detect clinical cases of depression and anxiety. Can the authors report and expand these considerations in a discussion section? (I suggest adding the reference: Impact of music on anxiety and pain perception among men undergoing prostate biopsy: Synthesis of qualitative literature. Dell’Atti L. et al. Complement Ther Clin Pract 2021; 43:101330. doi: 10.1016/j.ctcp.2021.101330).
Response 2: Thank you very much for this comment. We read with interest the reference you suggested and included it in our discussion (mini-review to present the use of music on anxiety and pain during prostate biopsy in ambulatory care setting) as it refers to the fact that yet there is no other study which yet examined the effects of audio- and audiovisual on patients receiving PPB. (lines 242-243). Ref (31) Dell'Atti L. Impact of music on anxiety and pain perception among men undergoing prostate biopsy: Synthesis of qualitative literature. Complement Ther Clin Pract. 2021;43:101330 (lines 330-331)